# Developing a Preference Scale for a Bear: From “Bearly Like” to “Like Beary Much”

**DOI:** 10.3390/ani13091554

**Published:** 2023-05-06

**Authors:** Jennifer Vonk

**Affiliations:** Department of Psychology, Oakland University, 654 Pioneer Drive, Rochester, MI 48309, USA; vonk@oakland.edu

**Keywords:** black bear, Ursus americanus, conditional discrimination, welfare, rating, ranking

## Abstract

**Simple Summary:**

I trained an American black bear in human care to choose different response buttons when presented with an image of either a highly preferred or a less preferred food item. The bear learned to choose the appropriate response button when presented with the preferred food item at above chance levels and differentiated between the use of the buttons appropriately. However, she did not reach a high level of performance with the less preferred food item even after over 1000 trials, suggesting that performing a conditional discrimination on the basis of preferences may be challenging for black bears. However, the work presented here represents the first attempt to train a bear to indicate her relative preferences using something like a Likert scale commonly used with humans to indicate their preferences and could be a valuable welfare tool for animals in human care. Similar work with gorillas suggests that bears are as capable as great apes in learning such tasks and would also benefit from this type of technical enrichment.

**Abstract:**

A preference scale for use by nonhuman animals would allow them to communicate their degree of liking for individual items rather than just relative preferences between pairs of items. It would also allow animals to report liking for images of objects that would be difficult to directly interact with (e.g., potential mates and habitat modifications). Such scales can easily be presented using touchscreen technology. Few zoos have used touchscreen technology for species other than nonhuman primates. I present a description of efforts taken to create such a scale for use with a single zoo-housed American black bear (*Ursus americanus*). Although the bear did not reach a high level of proficiency with assigning preferred and non-preferred food items to categorical responses of “like” and “dislike,” she was able to learn how to use the like and dislike buttons differentially for a single preferred and less preferred food item and she selected the correct response button for the preferred item at above chance levels. These data contribute to our limited understanding of black bear cognition and suggest that conditional discriminations may be difficult for black bears. This finding can inform continued efforts to create a simpler tool for nonhumans to communicate their preferences to human caregivers in a more nuanced way than is currently possible. More generally, the current study contributes to the growing body of work supporting the use of touchscreen technology for providing enrichment to less studied species like bears.

## 1. Introduction

While touchscreens are becoming increasingly common for enrichment or research purposes in zoo-housed nonhuman primates, the number of other species provided with this level of technical enrichment remains extremely small. Bears are widely recognized to be highly intelligent and curious animals that could benefit from more complex and dynamic enrichment. Although there is a paucity of work describing bears’ visual abilities, early work suggested that black bears discriminated various hues from grey, having difficulty with only red–green discriminations [1]. In addition, bears have been successfully trained to make categorical discriminations between stimuli presented on touchscreens (e.g., brown bears, *Ursus arctos*, Bernstein-Kurtycz et al., personal communication; American black bears, [2,3,4,5,6]; Malayan sun bears, *Helarctos malayanus* [7], and polar bears, *Ursus maritimus* (Jeremiasse et al., personal communication). Not only do bears appear to enjoy the stimulation provided from interacting with trainers through touchscreen training, the use of the computer presents researchers with novel ways to communicate with the bears. Computers have often been used in zoological settings to provide enrichment—most typically for nonhuman primates [8,9,10] but also for other species like parrots [11]—in the form of games, puzzles, or auditory enrichment. Computer interfaces have also been used to conduct assessments of animal well-being [9,12]. Computer interfaces can also be used to present images of foods and other objects, which subjects can then indicate their preferences for. For example, researchers recently presented a tablet to a Goffin’s cockatoo (*Cacatua goffiana*) so that the bird could select symbols representing various items, activities, or interactions. Their results suggested that the single cockatoo subject could use the tablet effectively to request objects and interactions that presumably had positive effects on her well-being [13]. The current study aimed to provide a means for a bear to symbolically communicate preferences for the first time.

Understanding individual preferences is critical for optimizing an animal’s environment and ensuring positive welfare. Preferences can inform habitat planning, husbandry, enrichment, and food provisioning [14]. Traditionally, preferences have been assessed indirectly by measuring degree of engagement with different enrichment items, foods, and environmental features etc. (e.g., [15,16]); objects that trigger different events like sounds [10], approach, and avoidance behavior (e.g., [17]); or efforts exerted to obtain access to space, social companions, or objects [18,19,20]. Preferences have been assessed more directly with forced choice tests between pairs or groups of real objects (e.g., [21,22,23,24,25]) or choices of symbols representing options like sounds [26]. These methods assess relative preferences among pairs or groups of choices, but fail to provide a more nuanced assessment of amount of liking (e.g., this object is liked to some degree compared to this item that is liked very much). Importantly, only objects that can be safely presented for investigation can be used in assessments involving real objects. In addition, paired-choice tests require the repetition of multiple pairings across items, which can be time consuming and can result in satiation when assessing preferences for foods or rewarding the individual for their choices. I attempted to develop a novel method to assess the degree of liking for various elements of the environment presented in pictorial form in an American black bear. This scale would ultimately allow her, and other nonhumans, to indicate preferences for food, enrichment, care staff, environments, sounds, and other stimuli that are not physically present at the time of assessment. Ultimately, I wished to be able to assess preferences for unfamiliar and previously inexperienced stimuli, such as planned habitat changes, possible mates, or even images representing more abstract concepts such as natural environments. Notably, preferences can be assessed in a single trial using this method once the animal understands the meaning of the end-points. I began training the bear to use such a nonverbal animal preference scale (NAPS) using images of foods for which her relative preferences could be determined.

In humans, preference scales are commonly encountered in product assessment, customer satisfaction surveys, and research into attitudes, beliefs, and personality traits. Such measures typically take the form of Likert scales [27], which allow respondents to indicate relative preference for items or agreement with ideas. One of the advantages of this type of scale is that respondents are able to indicate when they do not like an item at all rather than being forced to choose between equally preferred or non-preferred items. Rather, items are presented one at a time with a rating scale that has end-points representing a spectrum of agreement (e.g., from “strongly dislike” to “strongly like”). Using paired-choice tasks, an item might never be selected because it is less preferred than the other options, but it would not be possible to determine whether this item may also be liked rather than disliked. Preference scales have been widely adopted for research in multiple disciplines due to their flexibility [28]. Although there has been only one other known attempt to use such a scale with nonhumans, which I conducted concurrently with a bachelor group of gorillas (*Gorilla gorilla gorilla*, [29]), nonverbal versions have been used with human children [30,31] and clinical patients [32]. In these cases, verbal scale endpoints are replaced with intuitive images such as facial expressions to denote degree of liking [33], level of pain [34,35], and mood [36]. Therefore, although admittedly more complex and abstract relative to existing methods for assessing preferences, it seemed desirable and feasible to adopt similar methods to train nonhumans to use such a scale.

It should be noted that even pictorial Likert scales require verbal instruction, and a recent meta-analysis reveals that children below the age of five years cannot reliably use self-report measures of health outcomes. Furthermore, children below eight years of age may not be able to use a scale with more than two response options [37]. Therefore, training nonhumans to understand the construct of a sliding scale of preferences posed several challenges, not the least of which was deciding upon scale end points that might intuitively reflect “dislike” and “like.” Training animals to understand task requirements without verbal instructions is not a novel challenge, but does necessitate a prolonged period of training prior to administering the test in contrast to the type of one-off assessments conducted with human respondents. Second, because constructs of liking or preferences might rely on explicit self-knowledge, use of a preference scale may depend upon a degree of abstraction beyond the grasp of most nonhuman animals. There is no existing work that suggests that nonhumans can accurately report on their own (or others’) preferences when directly asked, and indeed, it has been noted that it would be difficult to learn signals of others’ preferences when they are discordant from our own, especially inconsistently so [38]. However, as a starting point, effective use of the scale could be acquired through a simpler process of association between items that evoke a particular visceral response (e.g., disgust, excitement) and different operational responses (use of the different response buttons). A process of generalization might support the appropriate use of response buttons associated with negative and positive feelings toward novel stimuli. Thus, use of the NAPS could be assumed to measure relative preferences for categories or objects that can be represented physically regardless of whether the subject explicitly represents the items as “things I dislike and things I like”.

Although stimuli to be rated could be presented in any modality perceived by the organism, use of a touchscreen system is most suitable for visual or auditory stimuli. Successful implementation of a visual NAPS requires that subjects understand the correspondence between pictures and their real-life referents. Many species have demonstrated picture–object correspondence (for review see [39]; e.g., in pigeons, *columbidae* [40]; in kea, *Nestor notabilis* [41]; macaques, *Macaca silenus* [42]), including the black bear that is the subject of the current study [2]. This apparently widespread ability supports the computer touchscreen methodology used here. However, another challenge with the NAPS is that stimuli, both in training and testing, must be subject-specific. To train subjects to use the NAPS, it is necessary to train them to understand what the different response buttons represent using stimuli for which the researchers already know the subject’s preference. These buttons must be presented at the extreme ends of a spectrum (i.e., spatially) so that responses representing intermediate levels of preference can be added later to allow a more nuanced scale of preference. Once appropriate use of the most extreme response buttons is established for items for which preferences are known, researchers can introduce the use of intermediate buttons, and finally, begin assessment of preferences for novel items. Here, I presented the bear with images of food items based on her preferences as indicated by her care staff to train her on the use of the scale.

I conducted a simple validation of the food preferences indicated by the care staff by presenting the bear with a set of images of preferred versus less-preferred food items on a touchscreen in a two-alternative forced-choice task. As with gorillas tested previously [29], it was expected that the bear would spontaneously select images of the preferred items. As expected, the bear selected the images of preferred over less preferred foods at above chance levels even when items belonging to the preferred and less preferred categories were continuously changed, which was done to ensure the generalizability of the concept. Others have provided a similar validation of the use of pictorial stimuli to assess food preferences in other species with many of these subjects showing generalization of choices to novel food images within the same categories of preference (e.g., sloth bears, *Melursus ursinus,* [43]; lion-tailed macaques, *Macaca silenus*, [42]; gorillas, [18,44,45]; Japanese macaques, *Macaca fuscata*, and chimpanzees, *Pan troglodytes*, [45]). Early work with black bears showing their stable food preferences [15] and my own previous work with black bears making natural category discriminations using a touch-screen (e.g., [4,6,46]) made me optimistic that this new black bear subject would become proficient in communicating her preferences for items presented visually using the NAPS. Furthermore, there is some existing evidence that great apes—at least those that have received some symbolic/language training—can appropriately use symbols representing “bad” and “good” [47], and can use pictures to communicate desires [48] and my previous work with bears and apes suggested that they were capable of representing similar levels of abstraction compared to great apes [3,4,5,6,46,49,50]. Ultimately, I wished to present the task with a 5-point scale, but given the difficulties of gorillas trained previously [29] with the use of a neutral response button, I began training the bear with only the two extreme (non-preferred and preferred) response buttons. Had she demonstrated proficiency with these two end-points, I would have gradually added in additional response buttons along the spatial continuum.

## 2. General Method

The studies reported here were approved by the IACUC of Oakland University (Protocol #12082) and the Animal Welfare and Management Committee of the Detroit Zoological Society.

### 2.1. Subject

One wild-born female American black bear (Migwan, Basel, Switzerland), age 11 years at the beginning of the study, participated in this study when she resided at the Detroit Zoo, Royal Oak, MI, USA. Migwan was rescued from the wild at a very young age and rehabilitated due to injuries. She was housed individually. Although experimentally naïve, Migwan had participated in husbandry training. For example, she was target trained using positive reinforcement, including clicker training.

### 2.2. Materials

All experiments were programmed in Real Practice or Inquisit 3.0 (millisecond.com) and presented on a Panasonic CF-19 Toughbook or an Asus Aspire One Laptop projected to a 19” VarTech Armorall capacitive touch-screen monitor. The touchscreen monitor was affixed to the front of a rolling LCD cart. The touchscreen monitor was secured flush to the front of the steel mesh with bungee cords to secure the screen in place so that Migwan could touch the screen with her tongue through the gaps in the mesh. The care staff member and researcher always tested the touchscreen from the bear’s side of the mesh prior to letting the bear into the indoor testing habitat. The laptop sat on a shelf on the cart behind the touchscreen (Figure 1). The experimenter stood against the back wall of the indoor area behind the cart and did not interact with the bear during trials. The care staff member placed the food rewards into a PVC tube affixed to the steel mesh to deliver food rewards for correct responses without any direct contact. This staff member always stood to the same side of the touchscreen during trials and did not direct attention to the bear or the laptop.

Stimuli used in the experiments were non-copyrighted photographs downloaded from various websites or images drawn in Microsoft Paint. These stimuli included images of various foods, and two-dimensional shapes such as circles, squares, triangles, and misshapen objects drawn in blue, yellow, red, and green. Food items used to reward correct responses composed a minimal proportion of the bear’s daily diet (e.g., almonds, biscuits, raisins, grapes).

### 2.3. General Procedure

The research took place in a non-public area of Migwan’s indoor habitat. She participated in testing three afternoons a week at around 13:00 h between April to September in 2014 and 2016. Migwan did not participate in testing from October to March as she was in a state of torpor during the colder months. During the spring and summer months of 2015, Migwan participated in other tasks including a picture–object correspondence test [2] and an ambiguous cue affective bias task [3]. She also participated in a novel judgement bias task that was conducted simultaneously from April to September 2016 [5]. Testing took about 10–15 min each test day, and Migwan completed 4–5 sessions of testing each day. Participation in the tasks was entirely voluntary. Testing for the day ended when Migwan had consumed an appropriate number of rewards as determined by the care staff. A flowchart of the experimental phases is presented in Figure 2.

If Migwan selected the correct stimulus, a pleasant auditory beep was emitted, the touchscreen turned white, and the care staff member assisting with the trials placed a small food reward down a PVC chute affixed to the mesh. If Migwan selected an incorrect stimulus, there was no audio feedback, the touchscreen turned black, and there was a 500 ms inter-trial interval.

### 2.4. Phase 1 Training

Migwan had already been trained to target and to station by her caretakers using positive reinforcement. Prior to beginning the study, she was trained by her care staff to station in front of the touchscreen without it being turned on. She was rewarded for targeting to a familiar target by touching it with her nose. Once she was reliably touching the target positioned right in front of the screen, the care staff removed the target and rewarded Migwan for touching the blank screen with her nose. This training took a period of approximately one week.

To train Migwan to use the touchscreen, I first presented her with a two-alternative forced-choice task where she was presented with two stimuli drawn in Microsoft Paint: a yellow square on a white background and a blue circle on a black background. She was reinforced for selecting the blue circle and not reinforced for selecting the yellow square. The idea was to create a positive association with the blue circle and not with the yellow square so that these would be intuitive response buttons for the end-points of the scale, with the blue circle representing “like or preferred” and the yellow square representing “dislike or less preferred.” The two stimuli filled most of the screen. The response button covered 80% of the stimulus so that Migwan had to touch the center of the stimulus and could not activate it just by nudging the edge of the stimulus. She was reinforced only if she used her tongue or nose to contact the touchscreen, not for using her paw. Migwan participated in, on average, four sessions a day, three days a week between April and July, 2014. The stimuli were presented in 20-trial sessions with the side of the correct stimulus (the blue circle) counterbalanced within the session.

In each trial, the stimuli appeared simultaneously and disappeared when one of them was selected. If Migwan selected the blue circle, a tone sounded, the screen turned white, the care staff member placed a food reward in the PVC tube affixed to the mesh, and the next trial commenced after 500 ms. If she selected the yellow circle, there was no sound, the screen turned black and she received no food reward. The inter-trial interval was the same. The criterion was set to four consecutive sessions at 80% correct or better (i.e., 16/20 correct responses) or two consecutive sessions at 90% correct or better (i.e., 18/20 correct responses).

### 2.5. Food Preference Assessments

To assess a spontaneous preference for images of preferred foods, I again used a two-alternative forced-choice procedure. Sessions included 20 trials and were identical to the training task described above except for the stimuli used. Migwan completed 39 sessions of this task. On each trial, a food indicated by the care staff to be preferred by Migwan was randomly paired with beets, lettuce, or carrots (on sessions 5 and 6), which were foods identified by care staff as being least preferred by Migwan. An image from the preferred category was randomly paired with an image from the non-preferred category on each trial and presented in random order. Table 1 indicates which food items were presented in each category on each session along with the number of trials on which each food image was presented within a session. One photo was used for each of these food types. Changes in the food items presented were made to test the generalizability of Migwan’s preferences. The side the non-preferred foods were presented on was counterbalanced within sessions with the constraint that they could not appear more than three times consecutively on the same side of the screen. Migwan was rewarded if she selected one of the presumed preferred foods and not if she selected the presumed non-preferred foods. 

### 2.6. Phase 2 Training Continuation

I next presented Migwan with a session of 20 trials of photographs of beets paired with the image of the blue circle (with side counterbalanced), where she was rewarded only for touching the blue circle to reinforce the idea of the blue circle as something positive and the beets as something not positive.

I then presented a single 20-trial session where the blue circle was paired with images of the preferred foods. As expected, she performed at chance, choosing the blue circle only 10 times, suggesting that she perceived the blue circle as equally positive, or likely to lead to reward, as the images of the preferred foods.

I then presented Migwan with five additional sessions of the blue circle paired with the yellow square to ensure she was still performing at criterion with the training stimuli.

### 2.7. Phase 3 NAPS Training

I created a computer program in Inquisit v. 3 that presented an image of a less preferred, or a preferred food in the center of the screen that, once touched, prompted the appearance of a response button in the top left (yellow square) or top right (blue circle) of the screen. The food image remained on the screen, centered in the bottom half of the screen once the response buttons appeared (Figure 3). Each response button took up about 30% of the top half of the screen. These sessions consisted of 10 trials (5 with beets and 5 with grapes). Only one image was used to represent each food type (beets for the less preferred item and grapes for the most preferred item) and it was the same image used during the food preference assessments described above. Migwan was trained to associate images of beets with the yellow square response button and images of grapes with the blue circle response button. On each trial, there was only one available response button. When Migwan selected that button, a beep sounded, the screen turned white and the care staff member placed a food reward in the PVC chute. The next trial began once Migwan had touched the image of the food and the subsequent response button with her nose or tongue. Migwan completed 7 sessions of this phase.

### 2.8. Phase 4 NAPS Training

In Phase 4 of Training, sessions consisted of 10 trials, which were the same as above except that both response buttons appeared simultaneously on every trial and Migwan was rewarded only if she chose the correct one (Figure 4). Migwan completed six sessions of this phase.

### 2.9. Phase 5 NAPS Training

Because I had successfully trained Migwan to associate the blue circle with reward, she was understandably reluctant to choose the yellow square even on trials when it would have been the correct response (i.e., when beets were presented). Therefore, I trained her to select the yellow square when beets were presented in this phase. This phase consisted of 10-trial sessions in which beets were always the food stimulus (always the same image as used previously) and both response buttons were presented simultaneously after she had touched the image of the beets (Figure 4). She was rewarded only for touching the yellow square/dislike button. Criterion was set to 80% correct responding for four consecutive sessions. Migwan received 13 sessions of this phase and then testing went on hiatus for fall torpor.

### 2.10. Testing Hiatus

When I resumed testing in April 2015, I focused instead on a picture–object correspondence task, which validated the use of two-dimensional images to represent objects for Migwan to rate [2]. I also presented her with a judgement bias test to assess affect changes across seasons [3]. In the spring of 2016, I returned to training Migwan in the current task. I trained her in several simpler conditional discrimination tasks using simple shapes (green triangle, red oval) rather than non-preferred and preferred foods to validate her ability to perform a conditional discrimination, before returning to the version of the task involving foods in the fall of 2016. Migwan learned to select a novel grey square response button conditional on being presented with a green triangle and to select a novel purple circle response button in response to being presented with a red oval. It took her 49 sessions (490 trials) to reach criterion and she successfully transferred at above chance levels to different images of the same shapes and colors as the original training stimuli. However, it took her 38 and 46 sessions to reach criterion again with the transfer shape and color stimuli, respectively.

### 2.11. Phase 6 NAPS Training

Having established that Migwan could learn this conditional discrimination task with less abstract decision rules, I returned to the task of training her to respond differentially to preferred and less preferred foods almost two years later. I presented Migwan with a version of the NAPS in which carrots were presented as the less preferred food and grapes were presented as the preferred food. I switched from beets to carrots as the less preferred foods to maximize the difference in appearance of the two presented foods as both beets and grapes were of a similar purplish color, and upon the suggestion of her care staff who noted that Migwan no longer preferred carrots relative to other foods from her daily diet. I verified that Migwan did not select carrots until all other foods were selected when presented with a handful of foods from her regular diet in her water trough. I also used the newly trained response buttons so that the dislike button was a gray square and the like button was a purple circle within a black background to mitigate against Migwan’s retained preference for the blue circle as the like button. The locations of the stimuli remained the same with the food appearing in the center of the bottom half of the screen and the dislike button appearing on the top left and the like button appearing on the top right.

Each session consisted of 10 trials: 5 in which carrots were presented and 5 in which grapes were presented, in random order. Both response buttons appeared simultaneously on the screen after the food item was selected by Migwan. She was rewarded for selecting the dislike button if carrots were shown and the like button if grapes were shown. Migwan completed 112 sessions of this phase between 4 August and 30 September 2016 before testing was halted. Testing took place three times a week at 13:00 h. Migwan simultaneously participated in a novel test of judgement bias during this time [5]. Migwan moved to another facility in 2017 and could no longer be tested.

## 3. Results

Analyses were conducted using SPSS v. 28. Alpha was always set to *p* = 0.05.

### 3.1. Phase 1 Training

Migwan reached criterion in 35 sessions (approximately 700 trials with some sessions missing some trials).

### 3.2. Food Preference Assessment

Initially, Migwan had a strong left side bias. She eventually met criterion with two consecutive sessions at 90% correct by session 33 but I continued testing her with additional minor changes to the composition of the food items and she reached criterion again with four consecutive sessions at 90% or better by her 39th session (780 trials). Overall, she chose the preferred foods at levels above chance determined by a one sample Wilcoxon signed rank test (*Z* = 4.434, *p* <0.001). Her performance improved across sessions, as can be seen in Figure 5. There was a significant difference in performance between the first and last halves of the testing sessions, Wilcoxon, *Z* = −3.732, *p* < 0.001.

### 3.3. Phase 2 Training Continuation

Migwan chose the blue circle 13 times on the single session in which it was paired with beets. In the single session, in which it was paired with preferred food items, she selected it 50% of the time. Across the five sessions in which it was paired with the yellow square, Migwan chose it on 80% or more trials on all but a single session, where she chose it 11 times.

### 3.4. Phases 3 and 4 NAPS Training

In Phase 3, only the correct response button appeared on each trial so Migwan was 100% correct on all 7 sessions. In Phase 4, Migwan chose the dislike response button only once across six sessions, so her performance was at chance.

### 3.5. Phase 5 NAPS Training

On the first four sessions, Migwan chose the dislike button correctly 50% of the time. However, by the end of 12 sessions, she had met the criterion, responding at 80% or more for four consecutive sessions. She was accidentally given a 13th session, on which she also scored 80% correct. Testing went on hiatus for torpor after this phase.

### 3.6. Phase 6 NAPS Training

Migwan completed 112 sessions. She responded equally quickly to touch photos of carrots (M = 2672.87, SD = 15,881.30) and grapes (M = 2596.05, SD = 18,051.841, *p* = 0.93 with a Wilcoxon signed ranks test).

Her average performance across all sessions was 56.61% correct, which was significantly above chance, (binomial test, N = 64, *p* < 0.001). She did not reach criterion; however, she had a run of three sessions at 80% correct between sessions 75 and 77 and she missed meeting criterion by a single trial by the 107th session. When comparing her performance on the first half of sessions (M = 55.36, SD = 1.33) to performance on the last half of sessions (M = 57.86, SD = 1.67), she showed little improvement. A Wilcoxon signed rank test confirmed no significant difference in performance between the first half and last half of sessions, *Z* = −0.123, *p* = 0.092. Figure 6 shows her performance across blocks of 4 sessions (40 trials).

Migwan’s performance on trials where carrots were shown was not significantly different from chance, M = 0.48, SD = 0.500, binomial *p* = 0.488, but her performance was significantly above chance when grapes were shown, M = 0.67, SD = 0.471, *p* < 0.001.

I also conducted Chi square tests of independence to test whether the food item presented on that trial was significantly associated with selection of the different response buttons. The likelihood of choosing a particular response button was significantly associated with the food that was shown, *X*^2^ = 26.066, *p* < 0.001. Migwan was more likely to choose the dislike button for carrots and the like button for grapes, as can be seen in Figure 7.

To test whether the latencies to respond were a function of the response button chosen (dislike, like, referred to henceforth as “response”) and correctness of the response (henceforth “correct”), I used a generalized linear model (GLM) with a gamma distribution and a log link function. I included response, correct, and their interaction as fixed effects in the model. Response significantly predicted response time, *X*^2^ = 34.092, *p* < 0.001, but correct did not, *X*^2^ = 0.139, *p* = 0.709. However, response interacted with correct to predict response latencies, *X*^2^ = 6.918, *p* = 0.009. The difference in response latencies for correct and incorrect responses was more pronounced if the dislike button was selected. In this case, Migwan was quicker to select dislike when it was the correct response (M = 952.99, SEM = 36.920) compared to when it was the incorrect response (M = 1082.40, SEM = 50.204). She showed the opposite pattern when choosing the like response. With the like response, she was faster to respond incorrectly (M = 1221.11, SEM = 45.816) than correctly (M = 1334.46, SEM = 43.864). These data appear in Figure 8.

## 4. Discussion

I report on the first attempt to train a bear to use symbols to communicate her preferences. Members of many other species, including nonhuman primates [47,51], domestic dogs [52], dolphins [53] parrots [54], and a cockatoo [13] have shown the ability to use symbols to communicate to varying degrees. Despite lofty intentions of training a black bear to use a touchscreen to communicate her preferences for two-dimensional stimuli, the task, which depended upon a conditional discrimination, proved very difficult to train, as it had been for three gorillas trained in parallel [29]. This was somewhat surprising as the bear had previously outperformed the gorillas in two conditional discrimination tasks used to assess judgement bias [3,5,49,50]. Furthermore, conditional discrimination tasks have been mastered by individuals of various species, including pigeons [55], rats [56], octopuses, cuttlefishes [57], squirrel monkeys [58], and chimpanzees [59], so the task should not have been beyond Migwan’s capability. Notably, the previous tasks involved associations between stimuli defined merely by shape and color and different response outcomes, whereas the current study aimed to test associations between broad abstract categories of preferred and less preferred foods. A construct concerning preferences is highly abstract and there is no existing evidence that nonhumans can represent a concept of their own or others’ preferences. Because the preferences of others can inconsistently match or differ from our own, it is likely a challenging construct for nonverbal organisms to represent [38]. However, there is some evidence that at least three language-trained apes appropriately used lexigrams representing “good” and “bad” and applied them in a manner that was appropriate and consistent with their human caretakers’ notions of good and bad behaviors [47]. There is also growing evidence that nonhuman mammals are capable of internally generating hedonic experiences in the absence of an external stimulus [60], making the use of a Likert scale for reporting preferences for symbolically represented aspects of their environment feasible. Furthermore, the current study progressed only to the point of training Migwan to associate a particular response with one preferred and one less preferred food, so should not have been conceptually more abstract than the previous studies. Unfortunately, due to Migwan’s move to another facility, I was unable to continue testing her. I had initially aspired to train Migwan to use a five-point scale indicating a more nuanced sliding scale of preferences for items for which I did not already know her preferences, but I was unable to reach this goal.

However, there are some promising data from this project. First, Migwan did learn to perform at above chance levels in the NAPS presenting only a single preferred and less-preferred food item, and she missed our somewhat arbitrary criterion level of performance by only a single trial. Thus, one could conclude that she acquired the discrimination. However, she performed with greater accuracy when presented with an image of the preferred food—grapes. This may not be surprising given that, in this final phase of training, I had replaced the previously trained less preferred beets photo with a photo of carrots. Carrots were less consistently presented as a member of the “less preferred food” category across all of the training presented here. One of the major limitations of this study is that I do not have data from systematic preference tests verifying the care staff’s indication of Migwan’s preferences, although I did conduct informal assessments by presenting multiple food items in the water trough and I observed that Migwan did not eat beets when presented as rewards and did not choose carrots when presented alongside other options.

When presented with forced-choice tests of preferred versus non-preferred foods, Migwan did not spontaneously select the images of preferred foods at above chance rates unlike two of three gorillas [44], lion-tailed macaques [42], and two sloth bears [43] tested in food preference assessments with images of foods. However, Migwan did choose the preferred foods at above chance levels across all 39 sessions, and did learn to select them to a criterion of 80% over the course of testing, suggesting that she may have formed categories for “preferred foods” over “less preferred foods.” It is less likely that she merely memorized which food photographs were associated with reward because I changed the food photographs periodically. However, it is true that beets, carrots, and lettuce were the only foods used as non-preferred foods so she may have simply learned not to select those images. I did not have the opportunity to test generalization to other photographs or to other preferred and less preferred foods to verify that she had formed such categories.

To further corroborate the conclusion that Migwan could learn to use the NAPS, she showed differential use of the response buttons dependent upon which food item had been presented on that trial, as did one of the gorillas tested in a similar procedure [29]. However, she used the buttons more accurately when the preferred grapes were shown, and did not clearly differentiate her use of the dislike and like buttons when the less preferred carrots were shown. It is possible that our initial training, in which I selectively rewarded Migwan for choosing the “like” button in order to reinforce its association with something positive, biased Migwan to the like button even when I changed its image in the final phase of training (the spatial location remained the same). It should be noted, though, that I did not train the gorillas selectively with the like icons used in their training, and they also struggled to learn this task with two of the three gorillas receiving many more trials in the initial training phase than the 1120 trials Migwan received [29]. Furthermore, although Migwan chose the like button more often, she did not choose it as often when it was incorrect as she did when it was correct. It was also not the case that every error involved inappropriate selection of the like button. Migwan also mistakenly chose the dislike button sometimes when grapes were presented. That she was above chance overall but did not show marked improvement across trials suggests that she had some spontaneous understanding of the task when I resumed testing in the fall of 2016, but did not develop an abstract conceptualization of the conditional discrimination nature of the task.

As with the gorillas who also struggled with this version of the task [29], I interrupted training on the NAPS with preferred and less preferred foods to present what I imagine to be a simplified conditional discrimination task using two-dimensional shapes of two colors (a green triangle and a red oval) associated with two new response buttons. Eventually, Migwan met a learning criterion. She met criterion more quickly with shape cues compared to color cues, which was in contrast to the gorillas who matched the colors more accurately. This is interesting because human children show a bias to attend to shape over color whereas chimpanzees tested in the same relational matching task showed the opposite bias and performed better on color matching trials [61]. These differences aside, both Migwan and two of the gorillas were able to learn the conditional discrimination task when presented with arbitrary shapes rather than images of food that were linked to their own preferences, suggesting that the mechanics of the task itself are not beyond their abilities, but that responding on the basis of their own preferences may be too abstract and require too much training to become a practical tool for use with animals in human care. This suggests that other procedures like token exchange [62] or the use of a progressive ratio reward schedule [63,64] to assess motivation to obtain rewards might have greater potential as a tool to assess animal preferences. In particular, although I had ultimately planned to present a 5-point Likert scale with five response buttons, adding more than two response buttons may be too challenging for nonhuman subjects [29], as human children cannot reliably use a 3-point scale until the age of eight years according to a recent meta-analysis [37].

Another limitation of the present study was the limited number of images used to represent the categories of preferred and non-preferred foods. Subjects show more robust transfer following training with a large number of exemplars representing categories, although training with multiple exemplars may slow acquisition of a category [65]. Training Migwan that she would not be rewarded for selecting the yellow square that represented the dislike button slowed her acquisition of the NAPS. In future, I would instead use images of items that held differential appeal for the like and dislike buttons. Although Migwan did quickly (i.e., within 120 trials) reach criterion in a task where selecting this button was always correct, she remained slightly biased toward the use of the like button even when the less preferred food image was presented and even when the images for the response buttons were replaced. That she did learn to use the dislike button, and to use it more often when it was rewarded (i.e., when the less preferred carrots were presented) indicates her flexibility in updating prior learned reward contingencies.

It is possible that presenting Migwan with other tasks in the intervening periods and breaking from training during torpor may have interfered with her reaching criterion levels of performance in this task. However, Migwan came very close to passing the admittedly somewhat arbitrary criterion. Furthermore, it should be noted that Migwan’s performance was quite exceptional in other tasks presented to her over the same period. In fact, she outperformed gorillas on several similar cognitive tasks [3,5,49,50]. Therefore, her ability to perform accurately was not generally hampered by the presentation of multiple tasks during the same period of testing. In fact, she demonstrated remarkable flexibility in switching between tasks.

## 5. Conclusions

Although this particular attempt to develop a NAPS suffered from several limitations, it is my hope that other researchers are inspired to improve on these methods. Developing such a tool would provide a valuable new method in which nonhumans could communicate their degree of liking for various items in a single trial, including for things that cannot be physically presented. The use of a NAPS, once trained, allows an assessment of stability of preferences over time without numerous repetitions of pairs of items over many trials. This may be especially appealing for assessing food preferences when foods cannot be presented repeatedly due to satiety or other factors. Notably, the basic idea for the NAPS can be extended to the presentation of auditory, olfactory, or tactile stimuli and response buttons can be presented in other forms other than touchscreen buttons. Thus, the basic paradigm could be easily modified to suit various species and modalities of presentation. However, use of the touchscreen allows for random presentation of stimulus items and the recording of both responses and latencies to respond. Both measures provided some indication here as to how Migwan was understanding the task. Although the present study was motivated by the desire to develop a novel welfare tool, it also provides some insight into black bear cognition. Bears are still quite understudied with regard to their cognition. Migwan’s performance in this study suggests that she can learn conditional discriminations, but that black bears, similar to other nonhumans, may not represent categories that are defined by unobservable features, such as relative preferences. Understanding such fundamental differences in how humans and nonhumans conceptualize their worlds will allow us to fully appreciate the uniqueness of other intelligent species and improve our abilities to provide them with the most appropriate stimulation while they are in our care.

## Figures and Tables

**Figure 1 animals-13-01554-f001:**
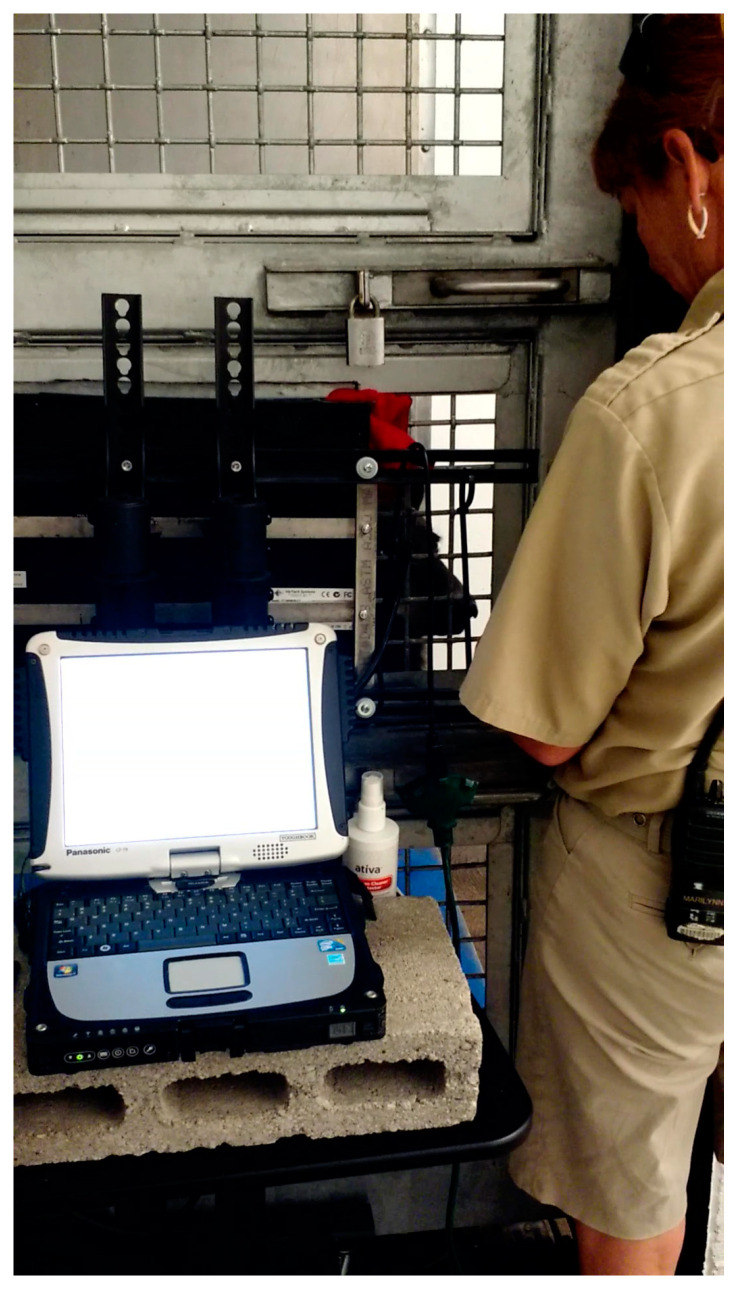
Experimental set-up showing Migwan peering from around the edge of the touchscreen.

**Figure 2 animals-13-01554-f002:**
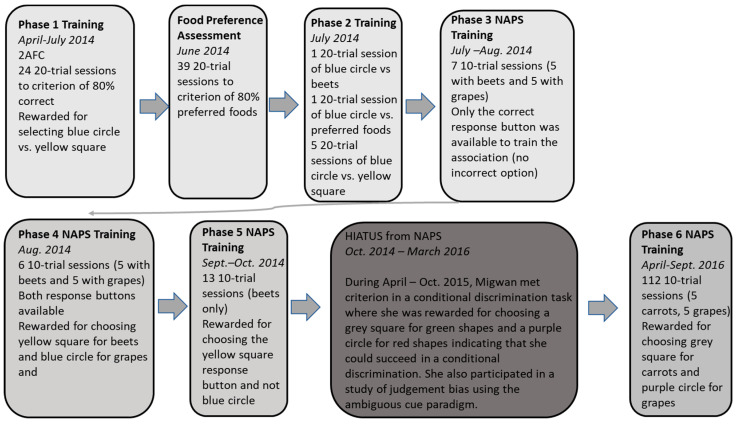
Flowchart of the phases of the experiment.

**Figure 3 animals-13-01554-f003:**
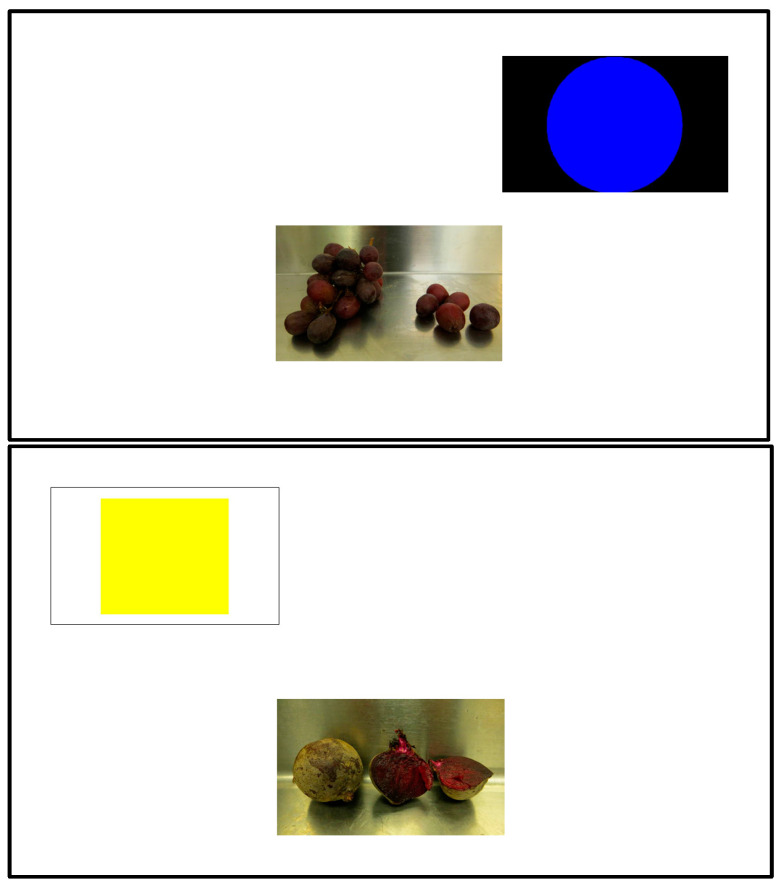
Sample trials of Phase 3 NAPS with the preferred food (**top**) and less preferred food (**bottom**).

**Figure 4 animals-13-01554-f004:**
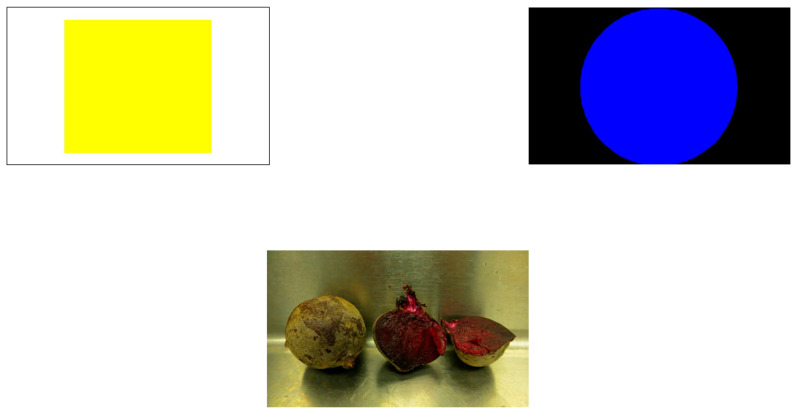
Trial of Phase 5 NAPS.

**Figure 5 animals-13-01554-f005:**
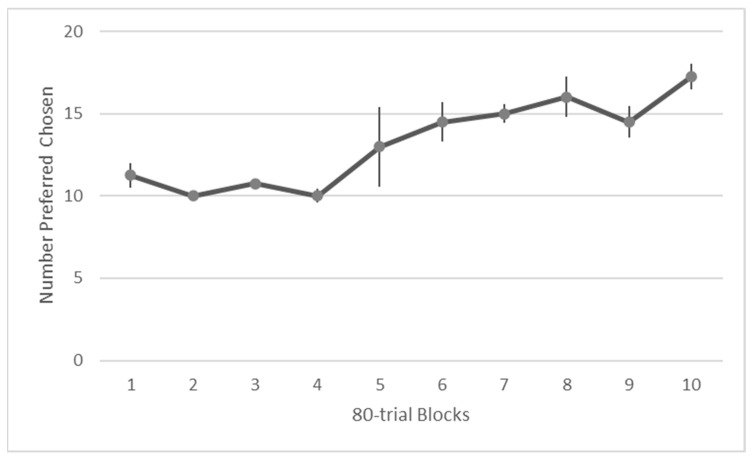
Average number of trials in which preferred foods were chosen during the food preference assessment.

**Figure 6 animals-13-01554-f006:**
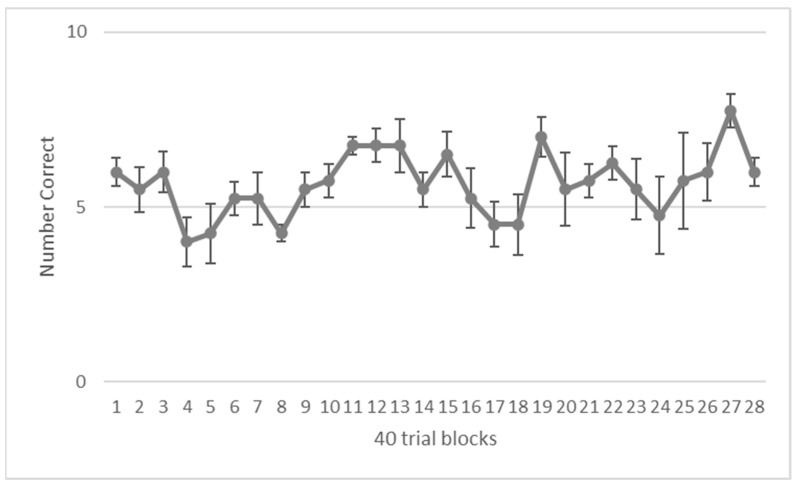
Performance on the Final NAPS Training Task.

**Figure 7 animals-13-01554-f007:**
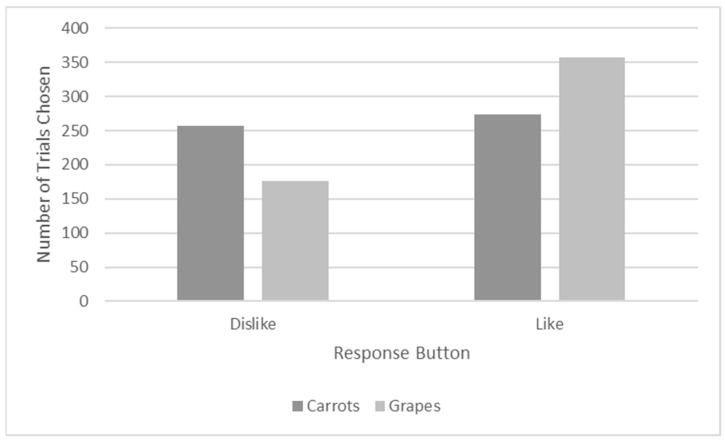
Number of Trials in which Responses were Selected for each Food in Phase Six.

**Figure 8 animals-13-01554-f008:**
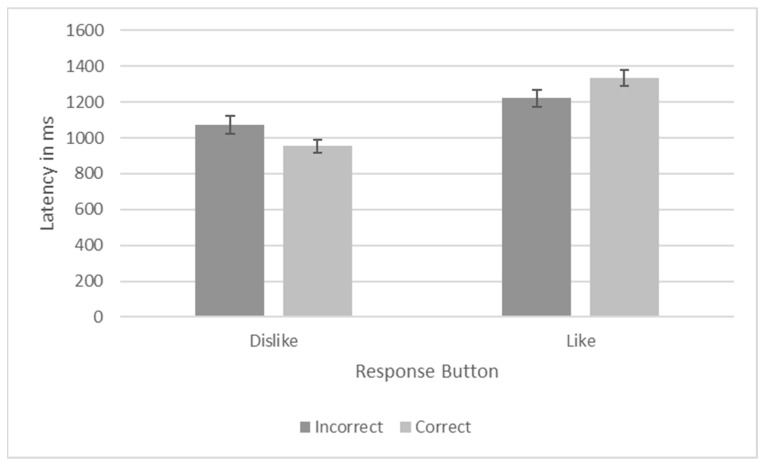
Response Latencies as a Function of Response and Correctness in Phase 6.

**Table 1 animals-13-01554-t001:** Foods presented for each session of Food Preference Assessment.

Sessions	Preferred (# of Trials in Parentheses)	Non-Preferred
1–4, 7	Apples (4), Sweet Potatoes (4), Grapes (4), Kiwi (4), Oranges (4)	Beets (20)
5–6	Apples (4), Sweet Potatoes (4), Grapes (4), Kiwi (4), Pear (4)	Beets (8) Lettuce (8), Carrots (4)
8–10	Apples (4), Carrots (4), Grapes (4), Kiwi (4), Oranges (4)	Beets (20)
11–13	Sweet Potatoes (5), Grapes (5), Kiwi (5), Pear (5)	Beets (20)
14–15	Apples (5), Pineapple (5), Pear (5) Oranges (5)	Beets (20)
16	Pear (5), Apples (10), Oranges (5)	Beets (20)
17–18	Apples (4), Sweet Potatoes (4), Grapes (2), Strawberries (2) Pear (8)	Beets (20)
19–21	Apples (5), Sweet Potatoes (5), Pear (5), Kiwi (5)	Lettuce (20)
22–24	Apples (7), Sweet Potatoes (6), Pear (7)	Beets (20)
25–27	Apples (5), Sweet Potatoes (5), Grapes (5), Kiwi (5)	Beets (20)
28–31	Apples (4), Carrots (4), Grapes (4), Pear (4), Cantaloupe (4)	Beets (20)
32	Apples (5), Orange (5), Grapes (5), Cantaloupe (5)	Beets (20)
33	Grapes (7), Pineapple (6), Sweet Potato (7)	Lettuce (20)
34	Grapes (4), Pineapple (4), Sweet Potato (4), Carrots (4), Kiwi (4)	Beets (20)
35–36	Grapes (5), Pineapple (5), Sweet Potato (5), Kiwi (5)	Beets (20)
37–39	Apples (5), Carrots (5), Pear (5) Oranges (5)	Beets (20)

## Data Availability

The data are available upon request from the author.

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
