# Peer review of "Developing a Preference Scale for a Bear: From “Bearly Like” to “Like Beary Much”"

_animals, 2023, doi:10.3390/ani13091554_

Round 1

Reviewer 1 Report

An interesting piece of work here with animals that have often languished in the zoo environment, lagging behind the enrichment levels we have seen in non-human primates despite their relatively equal intelligence being apparent. 

The use of technology in these situations is becoming more common, especially as both our understanding of species improves and the technology itself becomes more robust to allow interaction. Of course the limitation that is often apparent in zoo animal work is replication. Here we see the use of repeated trials within sessions to generate valid and testable data but this issue will always be challenging for more traditional scientists? Whilst I am not a biostatistician, the presented methodology seems to be in line with the pieces of already published work in the reference list and the visually presented data would support the statistical findings, in relation to the means and error bars, as far as my understanding of statistics goes.

I did find that the opening statements / abstract were maybe a little positive compared to the opening of the discussion and conclusion – I agree that there may be future development of this technique and training regimes but felt that it needed more reference to the data in the middle of the abstract.

The apparent complexity of the methodology with multiple phases was described reasonably well, but I don’t know whether tabulation or flow diagrams (for example used in other cited publications) might have aided the reader to see the acquisition of skills as the training progressed. For example see my minor comment below around line 220. I think this is where some aspects could be tightened up prior to publication to aid the reader (minor comment on line 330 for example linking the phases in the methods more closely to the results).

Whilst self-citation is commonly advised against (or limited) when publishing, the use of case studies of individual or small groups of animals makes this more likely and indeed maybe more acceptable. This is a relatively small field of study and I don’t find this to be excessive here given those facts.

Minor comments

Abstract (and line 132) – species name needs to be lower case. In addition, line 134 chimps get a Latin name but gorillas don’t.

Line 220 – I found this not as clear as it might have been, I assume that n=5 was the default position when 4 food items were presented, and exceptions are noted? The start point at session 5 and 6 – does that mean that the prior training was session 1-4?

Line 330 – how does the subheading “food preference” work into the various phases – does the figure here reflect all the phases? The subheadings are otherwise “phase” related but phase 2 is missing?

Line 614 – font change?

Author Response

An interesting piece of work here with animals that have often languished in the zoo environment, lagging behind the enrichment levels we have seen in non-human primates despite their relatively equal intelligence being apparent. 

The use of technology in these situations is becoming more common, especially as both our understanding of species improves and the technology itself becomes more robust to allow interaction. Of course the limitation that is often apparent in zoo animal work is replication. Here we see the use of repeated trials within sessions to generate valid and testable data but this issue will always be challenging for more traditional scientists? Whilst I am not a biostatistician, the presented methodology seems to be in line with the pieces of already published work in the reference list and the visually presented data would support the statistical findings, in relation to the means and error bars, as far as my understanding of statistics goes.

Thank you for these comments.

I did find that the opening statements / abstract were maybe a little positive compared to the opening of the discussion and conclusion – I agree that there may be future development of this technique and training regimes but felt that it needed more reference to the data in the middle of the abstract.

The sentence, “Although the bear did not reach a high level of proficiency with assigning preferred and non-preferred food items to categorical responses of “like” and “dislike,” she was able to learn how to use the like and dislike buttons differentially for a single preferred and less preferred food item and she selected the correct response button for the preferred item at above chance levels” was a concise synopsis of the primary data from this project. The other phases were really just training in support of getting to this testing phase so I did not go into detail on the results from these phases in the abstract. I did add a phrase, “These data suggest that conditional discriminations may be difficult for black bears but ….”

The apparent complexity of the methodology with multiple phases was described reasonably well, but I don’t know whether tabulation or flow diagrams (for example used in other cited publications) might have aided the reader to see the acquisition of skills as the training progressed. For example see my minor comment below around line 220. I think this is where some aspects could be tightened up prior to publication to aid the reader (minor comment on line 330 for example linking the phases in the methods more closely to the results).

All three of the reviewers have suggested the use of a flow chart so I have added one in the revision. It is true that the training with this bear was a bit messy. This is probably not uncommon for small N studies or case studies when piloting a procedure and making adjustments based on the individual subject’s responses, but it does challenge clarity of presentation. I hope the flow-chart aids in following the various steps of training. I have also attempted to create a table to simplify the presentation of the food preference assessment phase.

Whilst self-citation is commonly advised against (or limited) when publishing, the use of case studies of individual or small groups of animals makes this more likely and indeed maybe more acceptable. This is a relatively small field of study and I don’t find this to be excessive here given those facts.

I appreciate this acknowledgement. I like to be transparent about previous tasks that the subjects have participated in, as it can influence interpretation of their current performance, and of course the tasks used here were directly related to my published studies with the gorillas. To my knowledge, my own work is still the only published work with black bears using touch-screens, which led to some self-citations as this is the topic of the special issue on animal computer interactions.

Minor comments

Abstract (and line 132) – species name needs to be lower case. In addition, line 134 chimps get a Latin name but gorillas don’t.

This is because gorillas’ Latin name was already presented on line 84 and I was following the rule to cite only at first mention for each species. I have fixed the two cases where species names were capitalized. Thank you for catching this.

Line 220 – I found this not as clear as it might have been, I assume that n=5 was the default position when 4 food items were presented, and exceptions are noted? The start point at session 5 and 6 – does that mean that the prior training was session 1-4?

Yes, you assumed correctly in both cases. I have placed much of this information into a table instead in the revision in the hopes that it reduces confusion.

Line 330 – how does the subheading “food preference” work into the various phases – does the figure here reflect all the phases? The subheadings are otherwise “phase” related but phase 2 is missing?

Yes - the figure depicts data from all 39 sessions that were presented during this phase.

I apologize for the confusion. I did not report any data for Phase 2 here as I did not conduct any statistical analyses for that phase. I had simply verified that performance was as expected to justify moving on to the next phase as described in the methods. But I have moved the statements to results here to minimize confusion.

Line 614 – font change?

This has been corrected.

Reviewer 2 Report

An American black bear was trained on a food preference-based discrimination task on touchscreen equipment in the zoo setting. Results show that the bear could pick the appropriate button when preferable food was presented but not when the unpreferred food was presented. These results showed that the bear could not form abstract preferences. The author claims that the tool can be used to communicate the preferences of animals. Although the study would interest the general audience, there are major problems. The methods are explained in a very convoluted fashion, which makes the paper difficult to read. A graphical representation of the procedure is suggested (including the testing touchscreen setup). I suggest the author consider the signal detection theory approach to choice behavior and ex-Gaussian fits to the response times. 

Author Response

An American black bear was trained on a food preference-based discrimination task on touchscreen equipment in the zoo setting. Results show that the bear could pick the appropriate button when preferable food was presented but not when the unpreferred food was presented. These results showed that the bear could not form abstract preferences. The author claims that the tool can be used to communicate the preferences of animals. Although the study would interest the general audience, there are major problems. The methods are explained in a very convoluted fashion, which makes the paper difficult to read.

It is true that the training with this bear was a bit messy. This is probably not uncommon for small N/case studies when piloting a novel procedure and making adjustments based on the individual subject’s responses, but it does challenge clarity of presentation. I have tried to be as detailed as I can so that the study could be replicated or so that others could avoid some of the problems we encountered.

A graphical representation of the procedure is suggested (including the testing touchscreen setup).

I have included an image of testing.

All three of the reviewers have suggested the use of a flow chart so I have added one in the revision.

I suggest the author consider the signal detection theory approach to choice behavior and ex-Gaussian fits to the response times. 

Thank you for this suggestion but I am not sure how the signal detection approach would apply to these data where I do not have false alarms or misses. I have only correct/incorrect responses, which are two distinct responses associated with different target stimuli. That is, there are no failures to detect – there are simply correct or incorrect classifications. The main research question was whether the bear could learn to use the response buttons accurately (differentially depending upon the identity of the stimulus presented); therefore, the Chi square test of independence and comparison to chance seem to be the most appropriate ways to analyze these data.

Reviewer 3 Report

This is a very interesting attempt to provide a tool to probe nonhumans degree of liking for various items without repeating pairs of items in numerous trials. Following a previous study in 3 gorillas, the present study focuses on one female bear. The results are mainly negative but sharing negative findings is very important and Animals is a good venue for this paper. It is very well written and I really enjoyed the ambition and story. I have a few concerns/comments that I hope will strengthen the manuscript.

I found the idea of an easy to use likert scale for nonverbal animals, nonhuman or human, very attractive, but I would question its feasibility, at least in the discussion… Can nonverbal subjects actually reason "I like grapes, so if I see an image of grapes, I press the yellow square, and then I get a food treat (perhaps grapes, perhaps not) from my care giver", and the reverse for beets? The token task used in capuchins by Frans de Waal could be an alternative idea to follow, or the progressive ratio reward schedule used  to infer motivation in rats…

I would develop the section about the development literature in the Intro and would be explicit about the link between Likert scales and language. One has to understand verbal instructions to use a Likert scale, even a nonverbal one, based on emoticons for example. Likert scales might be adapted to toddlers not to preverbal infants and thus possibly not to animals see e.g. Coombes, L., Bristowe, K., Ellis-Smith, C. et al. Enhancing validity, reliability and participation in self-reported health outcome measurement for children and young people: a systematic review of recall period, response scale format, and administration modality. Qual Life Res 30, 1803–1832 (2021). https://doi.org/10.1007/s11136-021-02814-4

I would replace "unpublished data" by "personal communication".

Please indicate the nature and amount of reward given to the bear for each of the 80 daily trials or so, and which amount of food it represented in her daily food ration.

 Was the animal target-trained using positive reinforcement training ? In other words, was she familiar to an audio feedback such as a clicker sound before testing?

Was correction (representing the pair as many times as needed until the animal choose the other side) used to help Migwan move from spontaneously spatial responses to the requested object-guided decisions?

How many of Migwan' mistakes during the different testing phases were associated to her reverting to the spatial strategy?

Please provide a schematic overview of the experiment showing the stimuli used and the rule the animal had to apply during each training phase.

Based on a long experience in training laboratory macaques, a species that taught us a lot about learning, in cognitive tasks, Migwan's learning to nose-press the screen in about one week is very much like what a typical lab macaque would do (who, like zoo animals, are clicker-trained for daily activities). Once screen press is acquired and two items are presented, for macaques, like for us, choosing right over left (spatial discrimination) is spontaneous, choosing image A over B (object discrimination) is not their first choice but they learn it in a few daily sessions in particular if correction is used to help them. By contrast, to map a set of stimuli on a set of responses (if  image A then give response 1, if image B then give response 2, etc.) is called conditional associative learning, and is a real challenge to macaques who can take weeks to months to learn it (especially when the animal was trained with other rules before). The difficulty comes from the conditional rule which renders all possible responses correct, learning which response to avoid no longer does the trick… Training Migwan to associate images of beets with the yellow square response button and images of grapes with the blue circle response button is teaching her conditional associations, no walk in the park, even if one learns only this rule. So perhaps not the best choice  Rules for Migwan to apply were changed often for this experiment, which took place in the middle of other experiments, so I am not surprised she struggled. I would not expect a macaque to learn in these conditions either. Her difficulties could be due to too many successive rules possibly conflicting with each other.

 to provide a tool to probe nonhumans degree of liking for various items without repeating pairs of items in numerous trials. Following a previous study in 3 gorillas, the present study focuses on one female bear. The results are mainly negative but sharing negative findings is very important and Animals is a good venue for this paper. It is very well written and I really enjoyed the ambition and story. I have a few concerns/comments that I hope will strengthen the manuscript.

I found the idea of an easy to use likert scale for nonverbal animals, nonhuman or human, very attractive, but I would question its feasibility, at least in the discussion… Can nonverbal subjects actually reason "I like grapes, so if I see an image of grapes, I press the yellow square, and then I get a food treat (perhaps grapes, perhaps not) from my care giver", and the reverse for beets? The token task used in capuchins by Frans de Waal could be an alternative idea to follow, or the progressive ratio reward schedule used  to infer motivation in rats…

I would develop the section about the development literature in the Intro and would be explicit about the link between Likert scales and language. One has to understand verbal instructions to use a Likert scale, even a nonverbal one, based on emoticons for example. Likert scales might be adapted to toddlers not to preverbal infants and thus possibly not to animals see e.g. Coombes, L., Bristowe, K., Ellis-Smith, C. et al. Enhancing validity, reliability and participation in self-reported health outcome measurement for children and young people: a systematic review of recall period, response scale format, and administration modality. Qual Life Res 30, 1803–1832 (2021). https://doi.org/10.1007/s11136-021-02814-4

I would replace "unpublished data" by "personal communication".

Please indicate the nature and amount of reward given to the bear for each of the 80 daily trials or so, and which amount of food it represented in her daily food ration.

 Was the animal target-trained using positive reinforcement training ? In other words, was she familiar to an audio feedback such as a clicker sound before testing?

Was correction (representing the pair as many times as needed until the animal choose the other side) used to help Migwan move from spontaneous spatial responses to the requested object-guided decisions?

How many of Migwan' mistakes during the different testing phases were associated to her reverting to the spatial strategy?

Please provide a schematic overview of the experiment showing the stimuli used and the rule the animal had to apply during each training phase.

Based on a long experience in training laboratory macaques, a species that taught us a lot about learning, in cognitive tasks, Migwan's learning to nose-press the screen in about one week is very much like what a typical lab macaque would do (who, like zoo animals, are clicker-trained for daily activities). Once screen press is acquired and two items are presented, for macaques, like for us, choosing right over left (spatial discrimination) is spontaneous, choosing image A over B (object discrimination) is not their first choice but they learn it in a few daily sessions in particular if correction is used to help them. By contrast, to map a set of stimuli on a set of responses (if  image A then give response 1, if image B then give response 2, etc.) is called conditional associative learning, and is a real challenge to macaques who can take weeks to months to learn it (especially when the animal was trained with other rules before). The difficulty comes from the conditional rule which renders all possible responses correct, learning which response to avoid no longer does the trick… Training Migwan to associate images of beets with the yellow square response button and images of grapes with the blue circle response button is teaching her conditional associations, no walk in the park, even if one learns only this rule.  Rules for Migwan to apply were changed often for this experiment, which took place in the middle of other experiments, so I am not surprised she struggled. I would not expect a macaque to learn in these conditions either. Her difficulties could be due to too many successive rules possibly conflicting with each other.

Author Response

This is a very interesting attempt to provide a tool to probe nonhumans degree of liking for various items without repeating pairs of items in numerous trials. Following a previous study in 3 gorillas, the present study focuses on one female bear. The results are mainly negative but sharing negative findings is very important and Animals is a good venue for this paper. It is very well written and I really enjoyed the ambition and story. I have a few concerns/comments that I hope will strengthen the manuscript.

Thank you so much for your positive comments.

I found the idea of an easy to use likert scale for nonverbal animals, nonhuman or human, very attractive, but I would question its feasibility, at least in the discussion… Can nonverbal subjects actually reason "I like grapes, so if I see an image of grapes, I press the yellow square, and then I get a food treat (perhaps grapes, perhaps not) from my care giver", and the reverse for beets? The token task used in capuchins by Frans de Waal could be an alternative idea to follow, or the progressive ratio reward schedule used  to infer motivation in rats…

I share your skepticism having conducted the study. I do agree that it is probably too abstract for nonverbal animals, although one of the gorilla’s results from my related study do make me more optimistic that this could eventually be trained. In that study, one gorilla did generalize the appropriate use of the like and dislike buttons to novel foods based on his own preferences. If I had had more time to test Migwan in the task, I think she may have succeeded to generalize her use of the response buttons as well. But I appreciate your suggestion to consider advocating for the use of the other tasks mentioned and I have added this suggestion to the discussion.

I would develop the section about the development literature in the Intro and would be explicit about the link between Likert scales and language. One has to understand verbal instructions to use a Likert scale, even a nonverbal one, based on emoticons for example. Likert scales might be adapted to toddlers not to preverbal infants and thus possibly not to animals see e.g. Coombes, L., Bristowe, K., Ellis-Smith, C. et al. Enhancing validity, reliability and participation in self-reported health outcome measurement for children and young people: a systematic review of recall period, response scale format, and administration modality. Qual Life Res 30, 1803–1832 (2021). https://doi.org/10.1007/s11136-021-02814-4

Thank you very much for this reference. I have incorporated some discussion into the introduction as suggested:. “However, it should be noted that even pictorial Likert scales require verbal instruction and a recent meta-analysis reveals that children below the age of five years cannot reliably use self-report measures of health outcomes. Furthermore, children below eight years of age may not be able to use a scale with more than two response options (Coombes et al., 2021). Therefore, training nonhumans to understand the construct of a sliding scale of preferences posed several challenges; not the least of which was deciding upon scale end points that might intuitively reflect “dislike” and “like.” Training animals to understand task requirements without verbal instructions is not a novel challenge, but does necessitate a prolonged period of training prior to administering the test in contrast to the type of one-off assessments conducted with human respondents.”

I would replace "unpublished data" by "personal communication".

I have left this as is where I refer to my own work but have replaced the Jeremiasse reference as suggested.

Please indicate the nature and amount of reward given to the bear for each of the 80 daily trials or so, and which amount of food it represented in her daily food ration.

I have added, “On most trials, Migwan was given a small reward for a correct response, such as an al-mond or raisin or small cut piece of food from her daily diet. Therefore, the rewards amounted to a minimal proportion of her daily ration.”

 Was the animal target-trained using positive reinforcement training ? In other words, was she familiar to an audio feedback such as a clicker sound before testing?

I have added, “Although experimentally naïve, Migwan had participated in husbandry training. For example, she was target trained using positive reinforcement, including clicker training.”

Was correction (representing the pair as many times as needed until the animal choose the other side) used to help Migwan move from spontaneously spatial responses to the requested object-guided decisions?

No, we did not use correction trials. We found we did not need to as Migwan reached criterion in the basic conditional discrimination task with novel response button stimuli before we returned to testing after a more than yearlong break.

How many of Migwan' mistakes during the different testing phases were associated to her reverting to the spatial strategy?

I am not sure what you are referring to here by “spatial strategy.” Figure 5 shows how often she selected each response button in the final testing phase. In the previous Phase 5 Training phase, she was only shown the less preferred food, so all mistakes were to choose the like button.

Please provide a schematic overview of the experiment showing the stimuli used and the rule the animal had to apply during each training phase.

All three of the reviewers have suggested the use of a flow chart so I have added one in the revision. It is true that the training with this bear was a bit messy. This is probably not uncommon for small N studies when piloting a novel procedure and making adjustments based on the individual subject’s responses, but it does challenge clarity of presentation. I hope the new figure is helpful. Thank you for the suggestion to add the decision rule to be applied in each phase. I agree that will be helpful.

Based on a long experience in training laboratory macaques, a species that taught us a lot about learning, in cognitive tasks, Migwan's learning to nose-press the screen in about one week is very much like what a typical lab macaque would do (who, like zoo animals, are clicker-trained for daily activities). Once screen press is acquired and two items are presented, for macaques, like for us, choosing right over left (spatial discrimination) is spontaneous, choosing image A over B (object discrimination) is not their first choice but they learn it in a few daily sessions in particular if correction is used to help them. By contrast, to map a set of stimuli on a set of responses (if  image A then give response 1, if image B then give response 2, etc.) is called conditional associative learning, and is a real challenge to macaques who can take weeks to months to learn it (especially when the animal was trained with other rules before). The difficulty comes from the conditional rule which renders all possible responses correct, learning which response to avoid no longer does the trick… Training Migwan to associate images of beets with the yellow square response button and images of grapes with the blue circle response button is teaching her conditional associations, no walk in the park, even if one learns only this rule. So perhaps not the best choice  Rules for Migwan to apply were changed often for this experiment, which took place in the middle of other experiments, so I am not surprised she struggled. I would not expect a macaque to learn in these conditions either. Her difficulties could be due to too many successive rules possibly conflicting with each other.

I completely agree with you that it is likely that the conditional association nature of the task was challenging both for Migwan, and for the gorillas I trained in the same task. However, Migwan did learn another conditional rule in just 90 trials during the same period of time! (Vonk et al., 2021). In that task, a different spatial response was correct depending upon the background color of the screen. I think the difference in learning was due to the abstract nature of the rule to follow (a food I like or dislike) compared to “a green triangle or a red oval” which she also learned in a single season of testing. However, we trained her with only a single preferred food and a two less preferred foods, so the task presented here might have also been purely perceptual for her as well. Because she did perform above chance consistently without clear evidence of learning, I think a bias toward the right-side button due to earlier training strategies may have contributed to the lack of criterion level performance here.

I understand the more general concern about testing her on many different tasks and rules in a short period of time, but Migwan’s performance was exceptional in some of the other concurrent tasks so it was not the case that she was generally overwhelmed and confused about what was expected of her. In fact, she succeeded in the version of the conditional discrimination task using only green and red shapes over the spring and summer of 2015 when she simultaneously learned an ambiguous cue paradigm. Furthermore, she quickly learned another type of conditional discrimination (with a spatial outcome) in the spring and summer of 2016 while we were training her on the final task presented here.  But I do discuss the possibility of task interference in the limitations section around line 535.

Round 2

Reviewer 1 Report

I thank the author for their comments and the revision of the manuscript. I believe that the inclusion of the figures/tables aids the clarity of the methodology and results here. 

Author Response

I thank this reviewer for taking the time to provide a helpful review of my work.

Reviewer 2 Report

There are ways to apply SDT to 2AFC data but this is not a big issue at this point.

Author Response

I thank this reviewer for taking the time to provide a helpful review of my work. I would be interested in looking into this approach in my future work but, for the purpose of this special issue on animal-computer interactions, I want to focus more on the training steps and overall goals of the task rather than the measure of performance, as the testing phase was ultimately cut short. 

Reviewer 3 Report

The author adressed my concerns. I think the paper can be accepted.

Author Response

(The authors gave the same response as above.)
